# Metropolitan age-specific mortality trends at borough and neighborhood level: The case of Mexico City

Karol Baca-López[1,2☯], Cristóbal Fresno[3☯], Jesús Espinal-Enríquez[2], Miriam V. Flores-Merino[4], Miguel A. Camacho-López[1], Enrique Hernández-Lemus[2,5]*

1 School of Medicine, Autonomous University of the State of Mexico, Toluca, State of Mexico, Mexico,
2 Computational Genomics Department, National Institute of Genomic Medicine, Mexico City, Mexico,
3 Technology Development Department, National Institute of Genomic Medicine, Mexico City, Mexico,
4 School of Chemistry, Autonomous University of the State of Mexico, Toluca, State of Mexico, Mexico,
5 Centro de Ciencias de la Complejidad, Universidad Nacional Autonoma de Mexico, Mexico City, Mexico

☯ These authors contributed equally to this work.
* ehernandez@inmegen.gob.mx

**Data Availability Statement:** All relevant data are within the paper and its Supporting information files.

## Abstract

Understanding the spatial and temporal patterns of mortality rates in a highly heterogeneous metropolis, is a matter of public policy interest. In this context, there is no, to the best of our knowledge, previous studies that correlate both spatio-temporal and age-specific mortality rates in Mexico City. Spatio-temporal Kriging modeling was used over five age-specific mortality rates (from the years 2000 to 2016 in Mexico City), to gain both spatial (borough and neighborhood) and temporal (year and trimester) data level description. Mortality age-specific patterns have been modeled using multilevel modeling for longitudinal data. Posterior tests were carried out to compare mortality averages between geo-spatial locations. Mortality correlation extends in all study groups for as long as 12 years and as far as 13.27 km. The highest mortality rate takes place in the Cuauhtémoc borough, the commercial, touristic and cultural core downtown of Mexico City. On the contrary, Tlalpan borough is the one with the lowest mortality rates in all the study groups. Post-productive mortality is the first age-specific cause of death, followed by infant, productive, pre-school and scholar groups. The combinations of spatio-temporal Kriging estimation and time-evolution linear mixed-effect models, allowed us to unveil relevant time and location trends that may be useful for public policy planning in Mexico City.

## Introduction

Identifying trends in Metropolitan Mortality (MM) is a challenging problem nowadays. Systematic approaches to discriminate the relevance of social, economic, demographic, educational, environmental or criminal factors in MM are matters of intense current research [1–3].

MM can be differentiated from rural mortality since the causes and risk factors are substantially different [3]. To develop a useful model that may explain the evolution in time and space

**Funding:** The author(s) received no specific funding for this work.

**Competing interests:** The authors have declared that no competing interests exist.

of MM, is crucial to have a comprehensive information regarding the aforementioned factors. However, missing data is a common problem in developing and emerging countries, particularly for small-scale spatial and temporal level measurements. Diverse limitations and restricted access to public health data to collect information at the neighborhood and monthly level, lead to the modifiable areal unit problem (MAUP). It implies that a bias is generated affecting statistical hypothesis testing due to the combination of two or more spatial scales. To address this problem there have been several alternatives such as: correct the variance-covariance matrix using samples from individual-level data; focus on local spatial regression rather than global regression; design areal units to maximize a particular statistical result; developing statistics that change across scales in a predictable way; Bayesian hierarchical models as a general methodology for combining aggregated and individual-level data, or, using simulated data.

One of the most useful variables to explain MM is the age group, since the causes associated to risk factors, disease incidence and other several related variables are substantially different between such groups. Seminal works regarding age-specific MM can be found elsewhere [4–6]. Often, these age groups are separated into non-overlapping sets as follows:

1. Infant ($x < 1$ years old)

2. Pre-school ($1 \leq x < 4$ years old)

3. School ($4 \leq x < 14$ years old)

4. Productive ($14 \leq x < 64$ years old)

5. Post-productive ($x \geq 64$ years old)

In this work, we refer to the productive group as the economically productive population. Post-productive group is referred as economically dependent from the productive population. In addition, age-specific mortality can be influenced by social determinants at the individual level and at national or state level [7, 8]. Individual mortality is influenced by personal-level characteristics such as genetics, socioeconomic status and education [9].

However, there is growing evidence that collective or regional disadvantages can also be good predictors of individual mortality and population level mortality [1, 10]. Example of these regional disparities can be classified into social, demographic and environmental factors. In a recent study, Gavourova and Toth (2019) described how environmental factors influence changes in preventable mortality and how they impact differ from district to district in Slovakia [2]. In another context, individual and regional characteristics have been simultaneously analyzed for cardiovascular disease, to elucidate the effects of their interaction with air pollution, psychosocial stress, adverse childhood experiences and neighborhood deprivation index [11, 12].

Regarding the description of spatial mortality trends, two main approaches are used: i) All-cause or ii) Cause-specific mortality studies. In the first case, spatial and temporal variability have been measured, using different levels of granularity [13–17]. For cause-specific mortality several models have been developed for diseases such as cancer, diabetes, hypertension, chronic obstructive pulmonary disease, cardiovascular disease, hepatitis C and HIV/AIDS [6, 14, 17–26]. As an example of cause-specific mortality, Dwyer-Lindgren, et al. (2017), studied the variations in life expectancy, mortality rates and years of life lost from 152 causes of death at the county and neighborhood levels by age group and sex. They mainly conclude that county level estimations mask important local differences (between neighborhoods) [15, 27]. Studies with these characteristics have been performed in developed countries which generally have large, accessible and almost complete databases, thus, there is no MAUP.

In developing and emerging countries, it has been described an exacerbated problem in health disparities due to poverty, environmental threats, inadequate access to health care and educational inequalities that may lead to higher mortality rates [4]. In most cases, the conducted studies about mortality rates and other health outcomes rely on data that is not documented or incomplete. The last can be due to the lack of economic resources from public institutions or data restrictions for research purposes.

Mexico City, is one of the most populated cities worldwide. Its economy, employment rates and health services have improved over the last decades. Unfortunately, these resources are unevenly distributed, mainly affecting three risk groups: children, the elderly and the poor [28]. To worsen this situation, exposure to environmental risk factors derived from urbanization has increased, with its associated negative health effects gaining attention in recent years [29–32].

To set some context of mortality causes and trends in Mexico City, in a recent study, Aburto et al. (2018) analyzed lifespan and preventable mortality for Mexico City and the other 31 states of the country. They described changes in age groups. Particularly, they reported an increase in Diabetes and heart diseased-related mortality rates [33]. Furthermore, beyond diseased related mortality causes, an important number of deaths in Mexico City corresponds to homicides. Aburto et al. conducted different studies for the periods 2000-2010 and 2005-2015 regarding the impact of homicides in life expectancy and lifespan inequality. The authors reported an increase in homicide mortality that surpassed positive outcomes from health care reforms oriented to promote life expectancy at the national level [34, 35].

Regarding health outcomes, Gómez-Dantés, et al. (2016) report the leading risk factors for children and adults, supporting the fact that public health reforms and interventions should vary according to the specific risk factor for each age group. To give some examples of the latter, diarrhoeal diseases, undernutrition and poor sanitation were the leading risk factors for children, meanwhile for adults, chronic diseases and violence resulted the highest risk factors [36].

Despite the above mentioned previous background, at the borough and smaller scales, health, socioeconomic, educational or environmental disparities for Mexico City have not been formally quantified. In general, quality of life indexes of the urban and rural blocks are drastically different. However, these differences may be larger between urban neighborhoods than between rural and urban blocks.

Recently, new estimations suggested a high degree of social backwardness (term coined to refer to the lack of advancement of a group relative to the average) [37]. In Mexico City, health disparities such as illness and mortality rates can vary between close neighborhoods, just a few blocks apart, within the same borough. Taking this into account, assessment of health outcomes such as MM, using borough as a measurement unit, may result inadequate. A similar behavior arises from considering the temporal component: coarse-grained measurements may mask the variant behavior of MM. Once again, the MAUP is a matter to take into consideration at both spatial and temporal scales.

It is known that temporal variations in environmental factors (air pollutants for example) and socioeconomic variations (socio-economic level among boroughs and neighborhoods) can lead to biased results when not having measurements at the appropriate scales. Thus, there is an urgent need to count on fine-grained data in order to better interpret spatio-temporal correlations in health outcomes. To the best of our knowledge, there is no previous report in which time lapses for measurements were taken into smaller scales than in a year's time. Nevertheless, both spatio-temporal data are available in Mexico City for MM. These data include, at the borough level, information for general mortality, age, gender and other descriptive variables from 2000 to 2016 in a year basis.

In this work, using available data for Mexico City, a two-level spatial and temporal description for MM was analyzed using geo-statistical interpolation. In this context, the spatial component is tackled at borough and neighborhood scale, whereas the time scale was used in a year and trimester levels. Thus, using a multilevel modeling for longitudinal data provides a finer description of these phenomena that may result in a more accurate spatio-temporal explanation of MM in Mexico City. This in turn may allow to better capture the multi-scale complexity of mortality patterns in large urban areas such as Mexico City aiming at improving integrative policy designs.

## Materials and methods

Regarding the data collection in Mexico, the National Institute of Statistics, Geography (INEGI, Instituto Nacional de Estadística y Geografía) [38], each ten years develops a country-level census of population. In that national survey, economic, demographic and social data of all citizens in Mexico are collected and stored in a public database. Hence, INEGI provides the core of socioeconomic and demographic data, used as input for those databases that we used in this work. In addition, the mortality data was obtained from the Secretariat of Health in Mexico City (Secretaría de Salud de la Ciudad de México: SEDESA) [39]. Detailed description of both databases can be found in the following subsections.

### Study area

The study area is the capital of Mexico, Mexico City which was known as the Federal District (Distrito Federal) until 2016 (Fig 1A). Mexico City belongs to the Valley of Mexico Metropolitan Area, the biggest metropolis in the central region of the country. The city is divided into 16 administrative boroughs (municipalities) and 2, 097 neighborhoods, according to the geo-spatial information, i.e., polygon shapefiles, obtained from the Geostatistic framework, December 2018 (Marco Geoestadístico, Diciembre 2018) [40].

The neighborhood polygons were combined by borough identifiers to create the corresponding borough regions (Fig 1B), in order to have a multilevel hierarchical description of Mexico city. Borough centroids, were obtained using *rgeos* R package [41] (Fig 1C). In addition, neighborhood centroids were obtained by *gCentroid* or *gPointOnSurface* for convex or concave polygons respectively. All maps were created using R-software libraries: `sf` [42], `rgeos` [41], `raster` [43], `geosphere` [44], `spacetime` [45], `sp` [46], `rgdal` [47], `ggplot2` [48], `cowplot` [49], `gridExtra` [50] and `ggspatial` [51].

### Mortality database

A mortality database was created from available open data obtained at the Secretariat of Health in Mexico City (Secretaría de Salud de la Ciudad de México: SEDESA) [39]. The database comprehends a 16 borough, age-specific mortality rates follow up complete records, from the year 2000 up to 2016 and can be downloaded from [52].

The age-specific groups include the following five non-overlapping age descriptors:

1. Infant ($x < 1$ years old)

2. Pre-school ($1 \leq x < 4$ years old)

3. School ($4 \leq x < 14$ years old)

4. Productive ($14 \leq x < 64$ years old)

5. Post-productive ($x \geq 64$ years old)

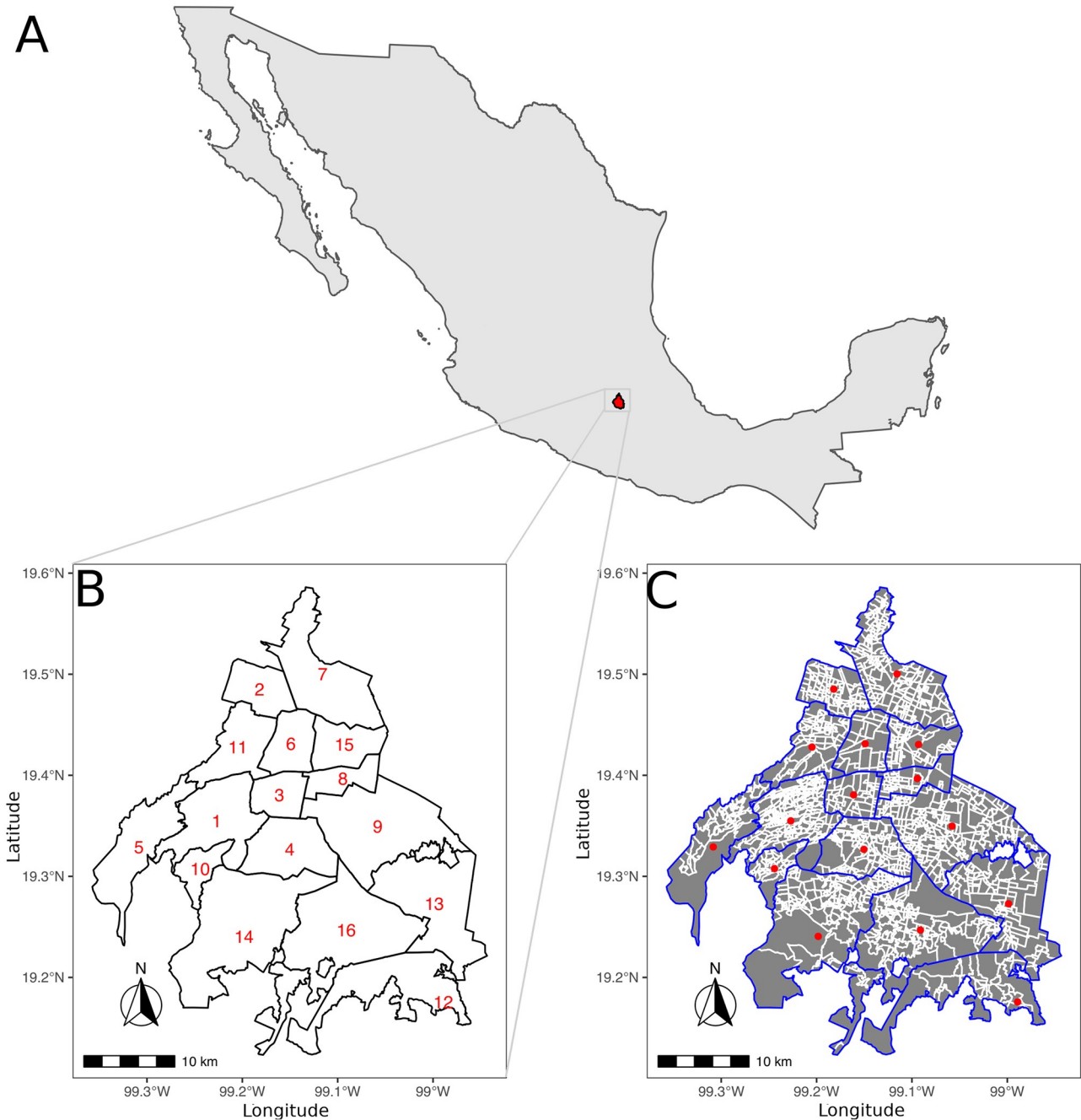

**Fig 1. Mexico City study area. A**) The map of Mexico shows the location of Mexico City (in red), formerly known as Distrito Federal, one of the 32 states of Mexico which is located in the central area. **B**) Borough level description of the 16 cases (in numbers) corresponding to: 1. Álvaro Obregón, 2. Azcapotzalco, 3. Benito Juárez, 4. Coyoacán, 5. Cuajimalpa de Morelos, 6. Cuauhtémoc, 7. Gustavo A. Madero, 8. Iztacalco, 9. Iztapalapa, 10. La Magdalena Contreras, 11. Miguel Hidalgo, 12. Milpa Alta, 13. Tláhuac, 14. Tlalpan, 15. Venustiano Carranza and 16. Xochimilco. **C**) Neighborhood level description. Blue lines describe borough limits, whereas white lines at neighborhood areas (in gray). Red dots depict borough calculated centroids. In panels A and B, the scale bar and north arrow are also included. Notice that some boroughs have a dense neighborhood description in comparison.

In this work, we refer to the productive group as the economically productive population. Post-productive group is referred as economically dependent from the productive population. The mortality rates are expressed, in all cases, per 1, 000 inhabitants, but the Infant rate, which was calculated per 1, 000 liveborn according to the data obtained from INEGI

and the Mexican National Population Council (Consejo Nacional de Población, CONAPO) [38, 52].

A five year time-step borough level evolution descriptive panel for every age-specific group is presented in Fig 2. Rows stand for age-specific mortality rate, and columns for a summarized time period (year) as reported by SEDESA. Naturally, with the available data at hand, the only alternative to visually explore the spatial data component, is to homogeneously color each borough with a single mortality rate for the reported year. Fortunately, there is discontinuity among boroughs, i.e., adjacent neighbor at using borough as area unit. Unfortunately, in this representation, neither time evolution nor spatial mortality age-specific rates are represented.

To overcome this drawback, a visual description at borough level is presented in Fig 3. It can be seen that there are different age-specific time evolution patterns (linear trend, in blue), such as a decreasing mortality rate for Infant and Post-productive groups, in contrast to the increasing trend in the Productive case. Although the Cuauhtémoc borough (in green) is the smallest one in terms of its neighborhoods (38 in total), it seems to overcome the mortality rates for most age-specific cases in comparison to the rest of the boroughs. No apparent time-dependent correlation can be observed for School and Pre-school, where several borough vibrant mortality curves are presented.

## Spatio-temporal interpolation

In order to have a more robust input dataset to analyze trends in MM, the measured variables must have a temporal component as fine-grained as possible, since it is well known that several risk factors associated with mortality have a cyclic behavior, grounded on temperature, air pollution, seasonal pathogens or even individual social aspects [53–55]. Several studies on MM have taken into account yearly data to associate certain variables with the response outcome [10, 20–22]. Often, monthly data are not available for developing countries; in other cases, a great number of missing data results very common.

In general, although different area-level health outcomes might share a variety of explanatory variables such as socioeconomic, pollution, delinquency levels, health access, among others; by down scaling, spatio-temporal heterogeneity might arise [20, 21, 56–58]. To overcome the MAUP, a variety of geo-statistical procedures have been implemented to estimate mortality rates at different granularity levels. For instance, Population-Weighted Average, local and global Empirical Bayes and Poisson Kriging have been used to estimate disease-specific mortality rates from age-adjusted data. Accounting for spatial correlation patterns for low and high frequency rates, Poisson Kriging have shown better results [59, 60]. Kriging methods allow to estimate spatial risk considering heterogeneity among small areas from poorly reported databases [61].

In this context, area-to-point (AtP) Kriging provides instantaneous estimation of the spatial regression, which is valid for each time point, i.e., it is appropriate for a cross-sectional study, nonetheless, we have both longitudinal and spatial data points. In addition, whenever a time-wise progression estimate is needed, AtP falls short of considering spatio-temporal correlations unless an appropriate time-regression procedure is considered, i.e., one compliant with Gauss-Markov theorem. In this case, a sum-integrated joint method—minimizing error via least squares regression—will give rise to a valid measure; formally a best linear unbiased estimator. Doing this, it will however be formally equivalent to a joint (sumMetric) integrated spatio-temporal estimator.

To overcome data scarcity, spatio-temporal Kriging has been used to jointly interpolate spatial missing data at the county level as well as for temporal interpolation in mortality data [6, 13, 62]. Kriging estimates has been also used to describe changes in child mortality trends,

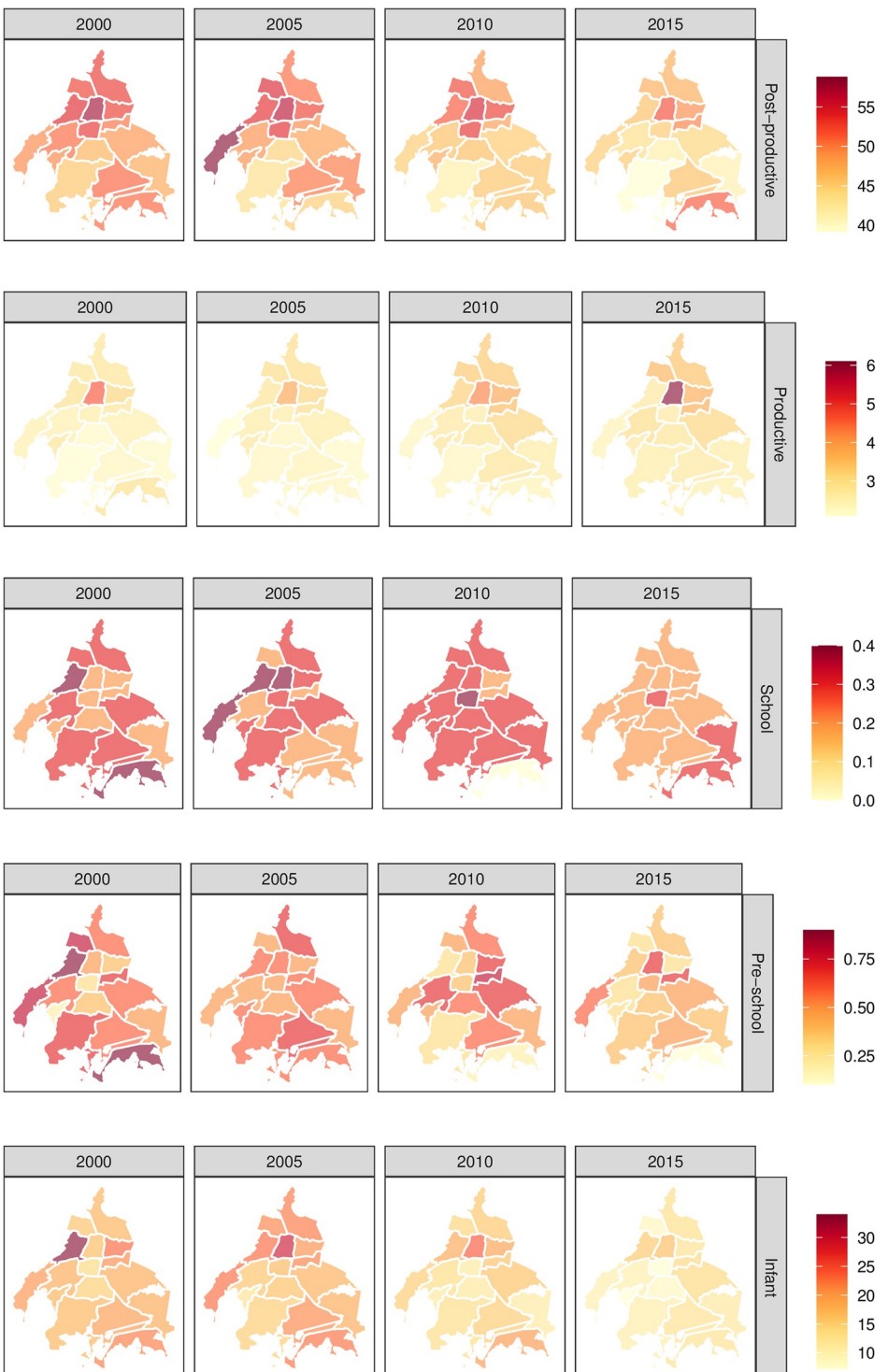

**Fig 2. Spatial age-specific mortality rates in Mexico City's boroughs.** Each row corresponds to a particular age-specific mortality rate, i.e., Post-productive ($x \geq 64$ years old), Productive ($14 \leq x < 64$ years old), School ($4 \leq x < 14$ years old), Pre-school ($1 \leq x < 4$ years old) and Infant ($x < 1$ years old). Each column stands for the selected years 2000, 2005, 2010 and 2015 from the total yearly available period 2000 to 2016. Mexico City boroughs are treated as a the unit area and color coded according to the corresponding mortality rate, which make them comparable by row. Interestingly, notice the different mortality rate ranges (color bars) depending on the age-specific group of analysis. Polygon shapefiles files can be freely downloaded at INEGI's website [40], whereas mortality data from SEDESA [52].

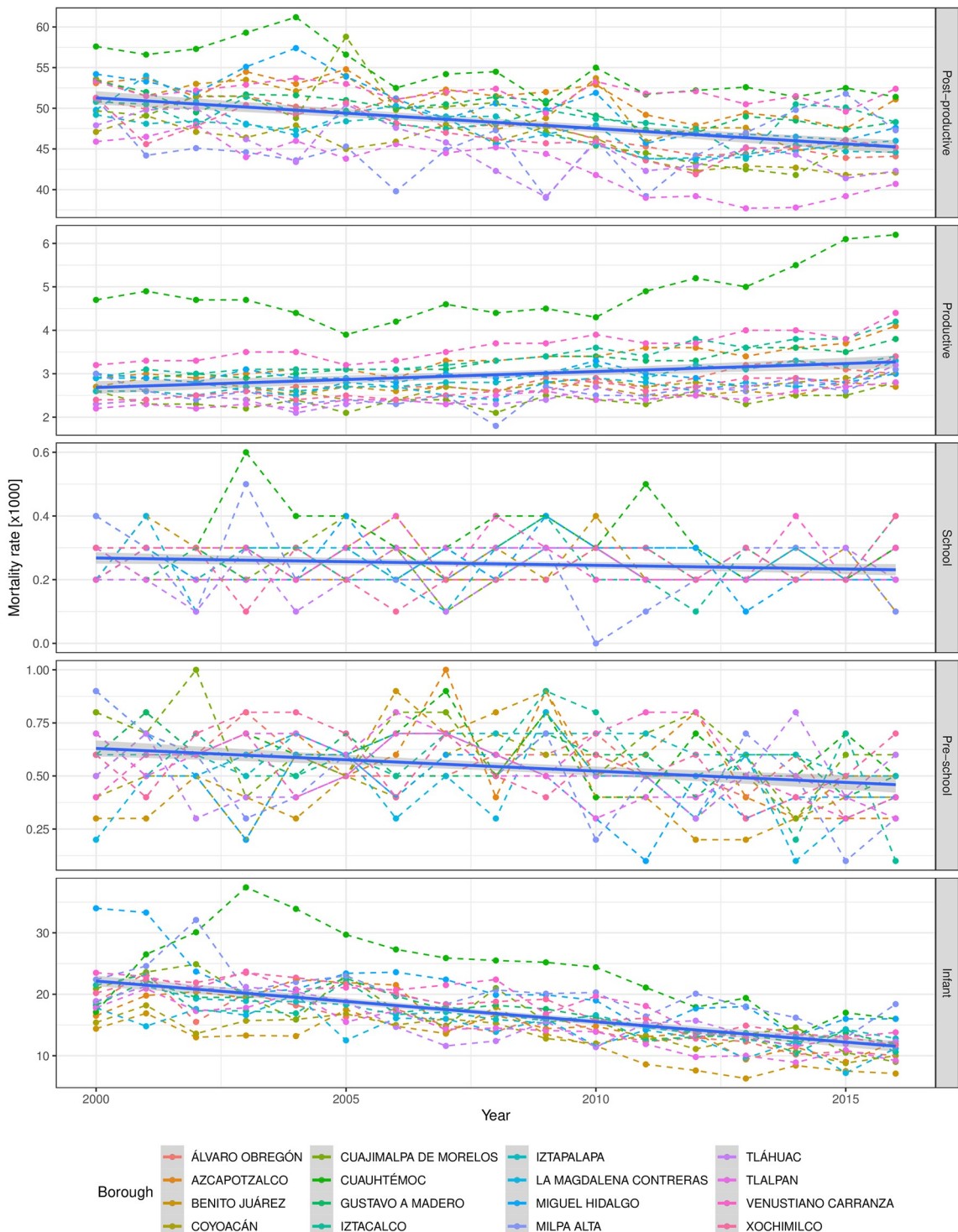

**Fig 3. Temporal age-specific mortality rates in Mexico City's boroughs.** Each panel corresponds to an age-specific mortality rate, i.e., Post-productive ($x \geq 64$ years old), Productive ($14 \leq x < 64$ years old), School ($4 \leq x < 14$ years old), Pre-school ($1 \leq x < 4$ years old) and Infant ($x < 1$ years old). All panels include complete age-specific mortality rates records by year from 2000 up to 2016 (in colour points) from SEDESA [52]. Color points stand for one of the 16 boroughs in Mexico City longitudinal mortality rate data. Borough data have been joined by their corresponding color dashed lines, whereas complete age-specific panel has been modeled by a linear regression (intercept and slope, blue line) with its respective standard deviation (grey area), to get a clear picture of the time evolution pattern. Interestingly, as the years pass, the mortality rate time evolution seems to diminish for Infant and Post-productive groups, whereas the Productive age-specific counterpart tends to increase.

evaluating between-countries and within-country sources of variation [5]. However, in current literature, a standard spatio-temporal methodology that properly addresses heterogeneity and data scarcity, has not been reported yet. The spatio-temporal description of the mortality database used in this work has been constructed on a yearly basis at the borough level.

Although, both spatial and temporal description levels can be accurate for open data summarized statistics, such description fails to represent the actual statistical metric unit, e.g., trimester measurements at the neighborhood level. In addition, it is known that in this context, data points have a spatio-temporal correlation which can be used to estimate values at unmeasured locations in space and time. Among the different geo-statistical interpolation methodological alternatives, the Kriging family provide unbiased estimates that minimize the variance-covariance spatio-temporal structure using an appropriate variogram modeling to tackle the MAUP [63]. Under this methodology, the choice of the variance-covariance spatio-temporal structure is not trivial and requires prior knowledge. As far as the authors know, we could not find in the literature reported mortality variance-covariance spatio-temporal structure. Thus, we tackle this issue by parsimony principle using computational force, in order to find the model structure with the least possible error among different mortality variance-covariance spatio-temporal structure available at `gstat` [64, 65] library: `metric`, `separable`, `productSum`, `sumMetric` and `simpleSumMetric`.

Briefly, a second data description level in both space and time, using neighborhood centroid locations and a trimester time basis, was carried out using Global Ordinary Kriging, for each age-specific mortality rate. All estimations were obtained using *gstat* R package implementation [64, 65]. Initial variogram parameter values were obtained from the complete mortality empirical (s)patio-(t)emporal rates variogram matrix $\gamma(s, t)$ (Table 1 in S1 File):

- **Nugget**: The median value of the first three empirical variogram matrix row/column means, for the spatial or temporal initial guess respectively.

- **Range**: The spatial range is one third of the lagged maximum spatial value; for the temporal case, it corresponds to the maximum value.

- **Sill**: The median value of the last five empirical variogram matrix row/column means, for the spatial or temporal initial guess respectively.

- **stAni**: The spatio-temporal anisotropy was estimated using a linear model as implemented in *gstat* R package [64, 65].

- **Joint spatio-temporal initial values**, are based on the mean of the independent spatial and temporal values, respectively.

Using the initial variogram parameters, different spatial, temporal or joint covariance structures were tested to find the best parsimonious correlated data description, according to the available implementations in *gstat* (*metric*, *separable*, *productSum*, *sumMetric* and *simpleSumMetric*) [64, 65]. All possible single, double or triple variogram combinations (Exponential, Gaussian and Spherical) were tested according to the corresponding covariance structure. Hence, computational power was used in order to find the best variance-covariance spatio-temporal structure using the appropriate variogram in order to use the parsimony principle. All covariance models were fitted using a quasi-Newton box constrained method, where only the lower-bound was set to 0.001 for every parameter. The upper-bound was left to its default value (infinite), i. e., no box constraint was imposed for the maximum value. The selection criterion to choose a covariance structure model, was to minimize the weighted mean squared error. Complete age-specific mortality tested covariance fitted model results can be found in Table 2 in S1 File. Finally, the spatio-temporal mortality rate interpolation was obtained using

all the available data points, under the best covariance model description, for the neighborhood centroids at a trimester level, in the period 2000 to 2016. Under this configuration the complete estimation root mean square error for the space-time Kriging data points was $8.16 \times 10^{-10}$ (see Table 3 in S1 File for age-specific errors).

## Time-evolution modeling

In the literature, there are many alternatives to assess the mortality time-evolution patterns using different types of models such as multilevel modeling for longitudinal data, longitudinal multilevel model, and longitudinal hierarchical linear model, among others. All of them target the different ways to model the variance-covariance structure, taking advantage of the inter/intra data structure and/or modifiable area unit.

Raudenbush & Bryk 1986 [66] took advantage of observed unit (individuals) when considering inter and intra school effects. Hence, their proposal can bee seen as a hierarchical or multilevel approach, where first, the within-group model is estimated by a separate regression equation for each school (referenced as Eq 1 in the original work). Then, the between-group model uses the regression coefficients as dependent output and try to model the within-school structural relationships (Eq 2 in [66]). Finally, the Raudenbush & Bryk can be coupled into a single equation by substitution of Eq 2 into Eq 1. The resulting equation allows to model the error term with many degrees of freedoms associated with the studied variables of interest.

However, the model estimation itself is not explained in detail. On the other hand, Kwok et al. 2008 [67] modeled variable time data points, but, they were focused on time correlation, hence, they introduce first-order auto-regression structure (AR(1)) to model the variance-covariance structure using SPSS (MIXED) and SAS (PROC MIXED) procedures. The two previous procedures are in essence, Linear Mixed-Effects Models. Recently, Anaya & Al-Delaimy 2017 [68] and Green et al. 2019 [69] did implement multilevel modeling for longitudinal data using linear mixed-effects models with R software [70] using the `lme4` package.

In order to assess the mortality contribution of the age-specific group, the spatial unit area (borough or neighborhood) and polynomial time evolution, we used a linear mixed-effects model to account for data constraints and lack of error independence using the definitions of Eqs (1)–(4) [71]. The model was specified using *Infostat* software version 2018, which is an R [70] front-end as follows [72]:

$$y_{ijk} = \mu + \alpha_i + \beta_j + \gamma \times t_k + \delta \times t_k^2 + \zeta \times t_k^3 + \alpha_i \times \beta_j + \alpha \times \gamma \times t_k +$$

$$\alpha_i \times \delta \times t_k^2 + \alpha_i \times \zeta \times t_k^3 + \beta_j \times \gamma \times t_k + \beta_j \times \delta \times t_k^2 + \beta_j \times \zeta \times t_k^3 +$$

$$\alpha_i \times \beta_j \times \gamma \times t_k + \alpha_i \times \beta_j \times \delta \times t_k^2 +$$

$$\alpha_i \times \beta_j \times \zeta \times t_k^3 + \varepsilon_{ijk} \tag{1}$$

$$\varepsilon_{ijk} = \lambda_{t_k} + v_{ijk} \tag{2}$$

$$\lambda_{t_k} = \phi_{\beta_j} \lambda_{t_{k-1}} + u_{t_k} \tag{3}$$

$$var(\epsilon_{ijk}) = \sigma^2 g^2(\alpha_i) \tag{4}$$

where, $y_{ijk}$ is the mortality rate for the *i*-th age-specific mortality group ($\alpha_i$), at the *j*-th borough or neighborhood level ($\beta_j$), for the *k*-th time ($t_k$); $\mu$ is the global mean; $\gamma$, $\delta$, $\zeta$ are the

corresponding third order time polynomial coefficients; in addition, the double and triple complete fixed effects model interactions and the error term $\varepsilon_{ijk}$ were also specified. Indeed, the error term $\varepsilon_{ijk}$ in Eq (1) is modeled in Eq (2) by a two-level model, to account for the lack of independent errors; where, $\lambda_{t_k}$ denotes the unobserved time effect and $v_{ijk}$ is the idiosyncratic error term. The correlated errors were tackled using a first order auto-regressive model for time dependence as described by Eq (3), where $\phi_{\beta_j}$ is the corresponding coefficient and $u_{t_k}$ the individual effect. Finally, the heteroscedasticity in Eq (4) was modeled as a multiplicative effect of the residual variance $\sigma^2$ times the variance error function $g(.)$ using a *varIdent* definition of the different age-specific mortality groups $\alpha_i$ [72].

The model was fitted using the R language with the `nlme` package under restricted estimation of the maximum likelihood [70, 73]. When possible, back-step model selection strategy was applied to remove the least significant fixed-effect term, one at the time, until no difference was found using a maximum likelihood test between competitor models. Type III sum of squares was used to assess an Analysis of Variance (ANOVA) table for marginal hypothesis tests for the fixed effects. Posterior Fisher's Least Significance Difference (LSD) tests were applied over statistical significant terms, using a multiple comparison Bonferroni p-value correction. When possible, bilateral test was used and Type I error was set to 0.05.

## Results

### Spatio-temporal variogram estimation

Regarding spatio-temporal age-specific mortality estimations, the initial guesses obtained from the sample spatio-temporal variograms are shown in Table 1 in S1 File (see Material and methods section). Depending on the age-specific mortality group, the initial guesses are different for the nugget, range, sill and spatio-temporal anisotropy (stAni).

In this context, the *nugget* is the model intercept attributable to the smallest error measurements or spatial sources of variation. Interestingly, these sources of variations are negligible for Pre-school and School, in contrast to the other age-specific groups, with a wide range of nugget values (0.01 − 21.31). In addition, the correlation extends between measurements, also known as *range*, in all cases is exactly the same for all age-specific groups and last about 12 years for as far as 13.27 km.

The variogram values obtained at the *range*, a.k.a. the *sill*, is as close to the *nugget* for the School and Pre-school age-specific groups, and departs from it at most to double its value for Infant group. The *anisotropy* remains the same for Post-productive, School and Infant groups, but differ for Productive and Pre-school counterparts. Final covariance model weighted mean square error for all the tested variogram permutations can be found in Table 2 in S1 File.

It is worth to mention that the lowest error for the different covariance structure methods included the *metric* case for School and Pre-school; *sumMetric* for Productive and Infant and *simpleSumMetric* for Post-productive was the best one. Within these covariance models, there was no apparent pattern in the winner variogram model tested permutation (temporal + spatial + joint). The Gaussian + Gaussian (Gau + Gau) was the winner's choice of temporal and spatial in the Post-productive age-specific group.

Moreover, Infant mortality also followed this pattern, with an additional Gau component for the joint variogram. The Exponential + Gaussian (Exp + Gau) was the winning choice for Productive mortality and Gau joint variogram. Indeed, this was a data-driven approach that required to explore the complete permutation grid, in order to reach a parsimonious spatiotemporal correlation model. A visual comparison of each winner covariance model and sample variogram can be found in File 1 in S1 File. Finally, under these configurations the complete

root mean square error estimation for the space-time Kriging data points was $8.16 \times 10^{-10}$ (see Table 3 in S1 File for age-specific errors).

## Two-level mortality rate spatial description in Mexico City

**Productive mortality rate.**   The spatio-temporal mortality description in Mexico City starts at the raw data presented in Fig 2. Let us consider the whole picture, using, for example, productive mortality rate as presented in Fig 4. Using the borough spatio-temporal granularity

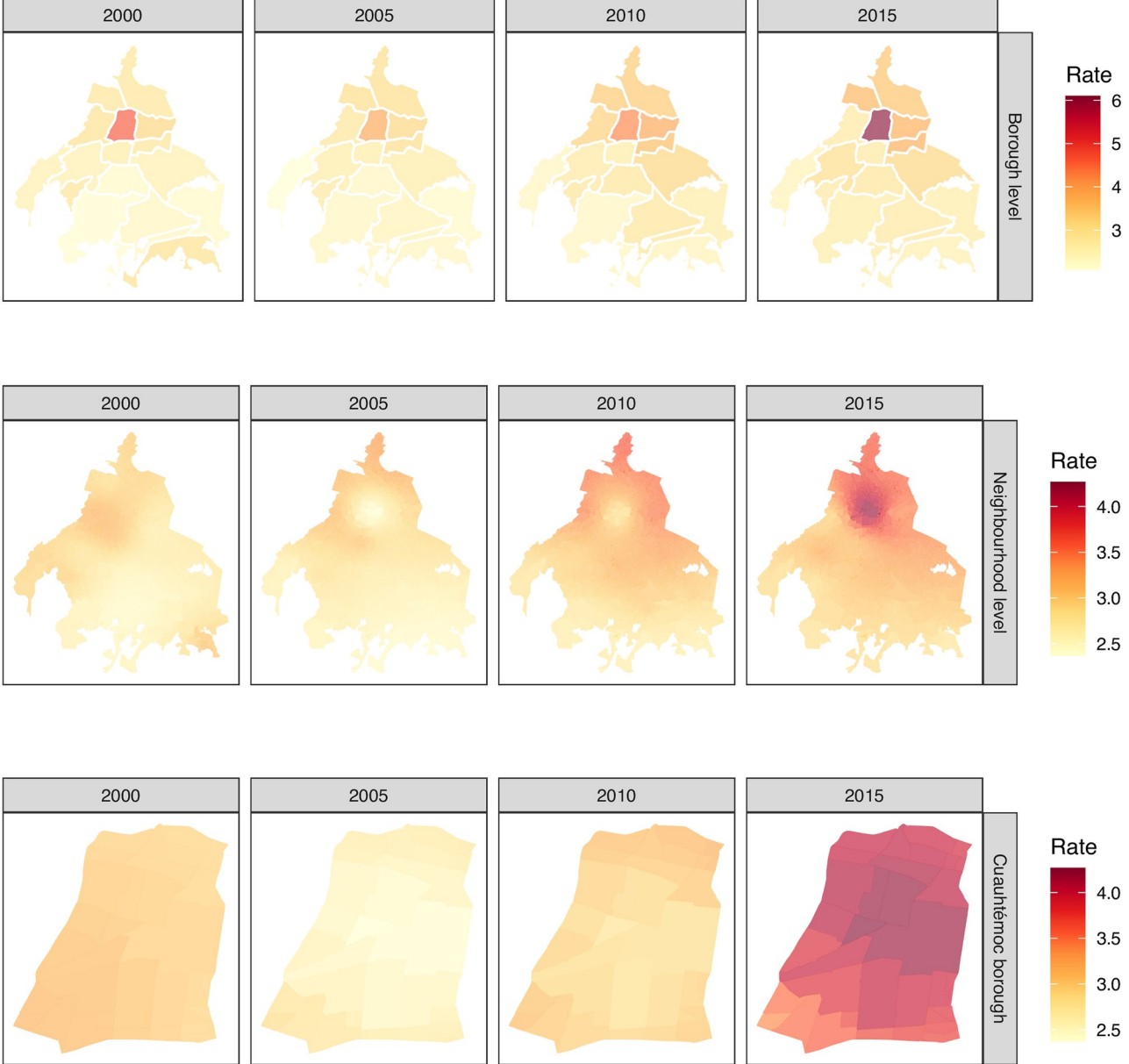

**Fig 4. Three levels of granularity for productive mortality rate in Mexico City.** First one, productive mortality rate original data description at borough level (panel **A**). Second one, productive mortality rate kriged data description at the neighborhood level is presented in panel **B**. In panel **C**, it is presented a zoom-in at a second description level for Cuauhtémoc borough. Interestingly, this is the borough with the highest productive mortality rate no matter the selected year (2000, 2005, 2010 or 2015), according to panel A and B (central borough in red). However, the mortality rate is not homogeneous at the neighborhood level, as depicted by the kriged values presented in panel C for the different years. Polygon shapefiles files can be freely downloaded at INEGI's website [40].

top panel in Fig 4, there is a clear spatial pattern, no matter the selected year (2000, 2005, 2010 or 2015). It seems that there is a global maximum mortality rate value (hotspot) at Cuauhté-moc borough (number six in Fig 1C). There, the mortality rate values radially decrease as long as we depart from this location to the outer boundaries of Mexico City. Indeed, the decrease is not homogeneous, i. e., it is dictated by a spatial anisotropy where the north and northeast direction have a less marked decline in comparison to the south and south-west direction. Moreover, at this temporal description, there is also an increment in the global maximum rates as we move from 2000 to 2015, in agreement with the temporal patterns presented in Fig 3 for the Global case.

Moving towards a deeper data exploration, the spatio-temporal interpolation obtained by Ordinary Kriging provides a productive mortality rate smooth surface (see neighborhood level at Fig 4). For a fair comparison, the same time scale was used (years), whereas the spatial description considered the neighborhood centroids grid. With this zoom-in, a more realistic geographical-continuous mortality transitions can be observed, unlike the discrete phenomena at borough level, for adjacent boroughs, presented at borough level in Fig 4. The latter reflects a far less abrupt change in mortality rates from one borough to another, thus giving a continuity between neighborhoods.

Furthermore, the productive mortality rates increase throughout neighborhoods and boroughs consistently from 2000 to 2015 in both spatial-scales. Also, notice that the mortality rate scale bar has also decreased from a maximum of 6 (at the borough level) into a 4.2 when we move towards into the fine grain spatial description for this age-specific group.

If we zoom-in even further, we can see the neighborhoods at a single borough, e.g. Cuauh-témoc borough Fig 4, where we can distinguish neighborhood level trends. Here, Cuauhtémoc borough was selected, since it has the highest mortality rate. Although apparently imperceptible, there are distinguishable differences in mortality values among Cuauhtémoc neighborhoods. Similar differences are obtained for the rest of the boroughs as seen at borough level description in Fig 4.

**School mortality rate.** Analogously to Fig 4, in Fig 5 we may observe the mortality rate through 15 years in the school group in three different levels of granularity. As it can be appreciated in the figure, mortality patterns in the school age are not homogeneously distributed in the 16 boroughs of Mexico City. Instead, each borough has its own pattern.

Unlike the previous case regarding productive age, in the school group the spatial trend is not clear. However, a visible and measurable decrease is observed from 2010 to 2015 in practically the whole city.

It is worth noticing that the scales in both figures are different. In Fig 4 the upper value for mortality rate is close to 6, meanwhile for school age, the top value mortality rate is around 0.4.

The comparison between those different groups in Cuauhtémoc borough and its neighborhoods is also remarkable. Meanwhile, for school group, 2015 is the year with the lowest mortality rate, that year was the highest for the productive age. Additionally, the differences in mortality rate at the neighborhood level is more visible in the productive group than the schoool one.

Similar two-level mortality rate spatial description in Mexico City can be found for the rest of the age-specific groups (Infant, Pre-school and Post-productive) in Figs 1-3 in S2 File respectively.

## Time-evolution mortality rate modeling in Mexico City

So far, we have filled the spatio-temporal gaps for the missing data points for both scales within the different mortality age-specific groups. Now, that we have overcome the summary time

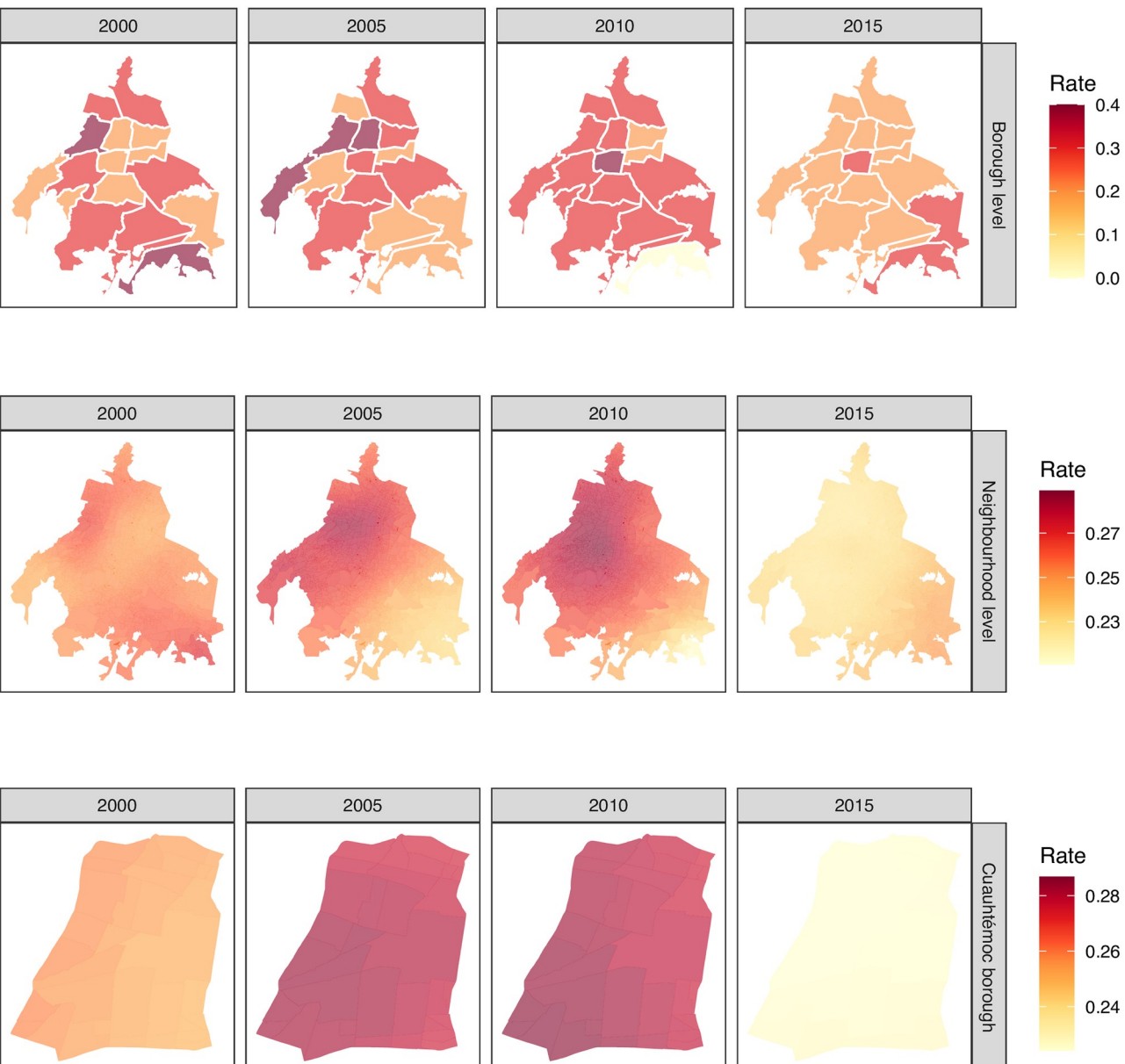

**Fig 5. Three levels of granularity for school mortality rate in Mexico City.** First, school mortality rate original data description is presented at borough level (panel **A**). Second, school mortality rate kriged data description at the neighborhood level is presented at panel **B**. In panel **C**, it is presented a zoom-in at the neighborhood level for the Cuauhtémoc borough. Interestingly, the school mortality rate for this borough increments from the year 2000 to 2005, while it reduces from 2005 to 2015 according to panels A and B, whereas the city's southeast area shows an increment from 2005 to 2015. However, the mortality rate is neither homogeneous nor constant at the neighborhood level, as depicted by the kriged values presented in panel C. Polygon shapefiles files can be freely downloaded at INEGI's website [40].

period for the mortality rate constraint, which is usually in a year base, we can study temporal/seasonal trends within the temporal range (12 years). In addition, we can explore the between mortality rate variability at borough level or even within its borough, a.k.a. neighborhood level data description. The later is a must, in order to address the MAUP, as we are changing the spatial scale, from several kilometers long (borough level), to a very different area unit scale at neighborhood (couple of blocks), where the prior estimated spatial correlation range is as far

**Table 1. Analysis of the variance at two spatial data description level.**

| Model Term | Borough level | | | neighborhood level | | |
|---|---|---|---|---|---|---|
| | Degs. of freedom | F-value | p-value | Degs. of freedom | F-value | p-value |
| $\mu$ | 1 | 17634.638 | <0.0001 | 1 | 281297.3649 | <0.0001 |
| $\alpha_i$ | 5 | 6981.1137 | <0.0001 | 5 | 6929.1962 | <0.0001 |
| $\beta_j$ | 15 | 8.3045 | <0.0001 | 33 | 6.5863 | <0.0001 |
| $t$ | 1 | 7.9808 | 0.0048 | 1 | 714.5003 | <0.0001 |
| $t^2$ | 1 | 34.1072 | <0.0001 | 1 | 248.0783 | <0.0001 |
| $t^3$ | 1 | 33.6751 | <0.0001 | 1 | 430.7367 | <0.0001 |
| $\alpha_i \times \beta_j$ | 75 | 20.0209 | <0.0001 | 165 | 5.6587 | <0.0001 |
| $\beta_j \times t$ | 15 | 2.9735 | 0.0001 | 33 | 9.0428 | <0.0001 |
| $\beta_j \times t^2$ | 15 | 2.4779 | 0.0013 | 33 | 2.2043 | <0.0001 |
| $\beta_j \times t^3$ | 15 | 1.9233 | 0.0177 | | | |
| $\alpha_i \times t$ | 5 | 4.172 | 0.0009 | 5 | 69.3982 | <0.0001 |
| $\alpha_i \times t^2$ | 5 | 7.2432 | <0.0001 | 5 | 88.1181 | <0.0001 |
| $\alpha_i \times t^3$ | 5 | 7.4925 | <0.0001 | 5 | 89.7661 | <0.0001 |
| $\alpha_i \times \beta_j \times t$ | 75 | 1.7383 | 0.0001 | 165 | 4.9837 | <0.0001 |
| $\alpha_i \times \beta_j \times t^2$ | 75 | 1.3908 | 0.0175 | | | |
| $\alpha_i \times \beta_j \times t^3$ | 75 | 1.1254 | 0.2224 | | | |

Type III sum of squares was used to assess the model defined in Eqs (1)–(4), where, $\mu$, is the global mean; $\alpha_i$ the age-specific mortality term; $\beta_j$ the borough or neighborhood term; $t$, $t^2$ and $t^3$ the third order time polynomial; and the double and triple interactions accordingly. These definitions were used at both spatial levels, i. e., borough and neighborhood. In addition, fixed effects, back-step model selection was carried out from the maximal to the current model at the neighborhood level. Empty cells correspond to discarded terms.

as 13.27 km. Finally, the mortality rate itself can be decomposed into both fixed-effects and random variance-covariance structure contribution using Eqs (1)–(4), according to the data level description as follows.

**Borough level mortality rate contribution.** Borough-level results are presented in Table 1. The ANOVA results showed that the only non-significant effect at the borough level, is the triple interaction that includes the time to the third power ($p = 0.22$). Hence, the mortality rate time-evolution pattern in Fig 3 can be parsimoniously captured by our methodological proposal. In addition, the auto-correlation parameter had an impact not as high as one would expect at borough level ($\phi = 0.12$). On the other hand, the variance function did address the different age-specific groups where Infant was the one with the highest value (11.59) followed by Post-productive (10.34), Pre-school (0.85), Productive (0.82) and School (0.43).

To further explore the mortality behavior, posterior Fisher's LSD tests results were obtained over the age-specific, $\alpha_i$, Mexico City's borough, $\beta_j$, and double borough times age-specific interaction terms $\alpha_i \times \beta_j$ (Fig 6A–6C, respectively). The first remarkable result is the mortality contribution to the main effects $\alpha_i$ and $\beta_j$. Mexico City's model results evidence, for the time-period studied, suggests that the age-specific term ($\alpha_i$) is the main responsible for the mortality rate when compared to the spatial borough contribution ($\beta_j$). Indeed, the mortality rate means are definitely empowered by age-specific groups (upper bounded at 50 [x1000]) rather than spatial borough locations, which is upper bounded at 17[x1000].

Secondly, the age-specific mortality rate groups do not overlap between each other, due to the different Fisher's LSD letters (A-E) in Fig 6A. Moreover, the Post-productive group (A) outruns any other group, but also doubles its following competitor, the Infant group (B). Interestingly, this trend also remains when we compare two consecutive groups, i.e., B vs C, C vs D and so on. In this age-specific context, the school is the group with letter E, i.e, is the one with

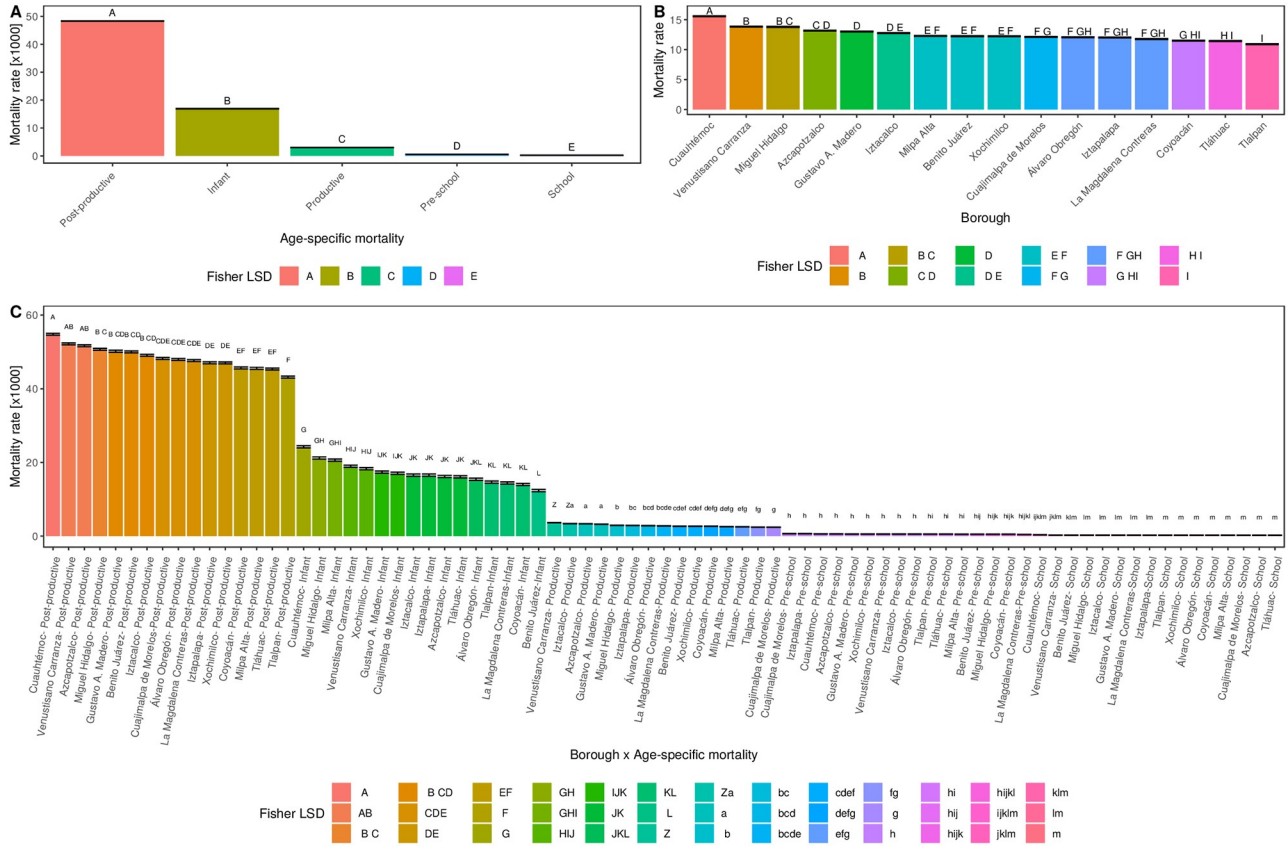

**Fig 6. Posterior mortality rate test results at borough level for Mexico City.** Fisher's Least Significant Difference (LSD) tests were performed over the estimated age-specific mean mortality rate, according to model description in Eqs ((1)–(4)). In panels, Mexico City Fisher's LSD test results for: **A**) Age-specific mortality, **B**) Borough level, and, **C**) Borough × Age-specific interaction. The LSD group mean test results are presented as bar-plots with their corresponding standard deviation bar and mean group letters (A, B, C . . .), where bars that share at least a single letter, are not statistically different after Bonferroni multiple test correction ($p > 0.05$). Interestingly, mortality rates is mainly composed by age-specific contribution when compared to Borough impact. In addition, Borough × Age-specific interaction retains the mortality age-specific rate pattern, but, it is modulated by the borough contribution.

the least mortality rate. Interestingly, previous results could be, to some extend, the explanation that models the Mexico's actual population pyramid shape data [74]—pencil like figure, i.e., long bar with sharp-pointed end.

Here, the Post-productive people are the most underrepresented group with a sharp-pointed shape at the tip of the population pyramid, due to the high mortality rate evidence of our borough level model results. Following this rationale, the second highest mortality rate group is B (Infant), which could be probably the reason why the typical pyramid shape is broken into a uniform/bar shape shared by the Infant, Pre-school, School and Productive age-specific mortality groups, influenced by their low mortality rate.

Thirdly, taking into consideration Fisher's LSD results for the 16 borough's mortality rate contribution $\beta_j$ (Fig 6B), it can be seen three important aspects: i) Among group letters from A to I, Cuauhtémoc (A) and Tlalpan (I) are the boroughs with the highest and lowest model estimated mean values respectively; ii) The LSD mortality rate group mean letters cluster up to five boroughs per cluster, i.e., borough that share a single letter, are not statistically different after Bonferroni multiple test correction ($p > 0.05$), thus, belong to the same cluster; iii) The borough Fisher's LSD cluster structure is correlated to spatial proximity.

Fourthly, moving towards $\alpha_i \times \beta_j$ group results (Fig 6C), i.e., age-specific times borough model's interaction LSD group means, it can be seen that the original and predominant age-specific mortality rate contribution in panel A is modulated by the borough contribution of panel B. Thus, if we average LSD group mean results in panel C, by age-specific groups, we should return into panel's A results. At this level, we are dealing at "between" borough mortality rate level description, where the number of Fisher's LSD group letters has increased proportional to the number of possible $\alpha_i \times \beta_j$ levels.

**Neighborhood level mortality rate contribution.** In order to model "within" borough mortality rate data description, we need to change our attention into another spatial scale representation, a.k.a. neighborhood level. If we pick the borough with the highest mortality rate (Cuauhtémoc), we found that the model presented in Eqs (1)–(4) is not well-suited for this level description data. Hence, a model selection process was considered in order to obtain the best parsimonious data description.

The current model results are presented in Table 1, where it can be seen that some cells are empty due to terms discarded from the analysis. Interestingly, the third and second order time triple interaction terms have been excluded from the analysis, in addition to the borough times time to the third power. Hence, in this context the modeling complexity has been reduced, at the expense of a higher autocorrelation ($\phi = 0.88$) and different variance function parameters and ranking, i. e., Productive (3.80), Post-productive (2.43), Pre-school (2.22), School (1.35) and Infant (0.04).

The Cuauhtémoc borough results can be found in Fig 7. Its 38 neighborhoods were numbered according to the high mortality rate downwards (Fig 7A). This result is complemented by Fig 7B, where the Fisher's LSD means were used to describe the mortality rate landscape. Interestingly, at the neighborhood level, there is also a radial mortality rate decay starting at the central neighborhood with number one and the letter A, which correspond to Tabacalera neighborhood. Conversely, the two neighborhoods with the lowest mortality rate (letter D) are situated at the opposite borough extremes—south-west (Hipódromo de la Condesa) and northeast (Valle Gómez) borders. Moreover, the age-specific mortality patterns have changed from Post-productive and Infant mortality at the borough description in Fig 6A into Post-productive and Pre-school as seen in Fig 7C. Moreover, the Fisher's LSD test results in this borough, but at the neighborhood level, showed some regions with up to four possible overlapping letters (Fig 7D).

Taking about the different mortality rate model description, we have to keep in mind that Fig 6 panel B "between borough" mortality rate data order of magnitude, is going to be fine grained ("within borough") modeled using the same framework depicted in Eqs (1)–(4). The first result to be discussed is that, unlike borough level, neighborhood mortality rate data does not cope with third order neighborhood interaction and the only triple significant interaction has a linear time tendency for age-specific times neighborhood mortality, as shown in Table 1.

The second aspect, is the new insight of the mortality rate at the neighborhood level. Here, the first mortality rate level description (at borough) leaves Cuauhtemoc's borough near 15 [$x$1000]. Now, Fig 7 panels C and D, decomposed into a finer grain considering the same age-specific groups, but, now Cuauhtemoc's neighbors are included, respectively. At this data level description, mortality age-specific groups within Cuauhtemoc are closer to each other (same magnitude order, units) unlike borough level (one magnitude order, tenths). In addition, between the neighborhood mortality rate is almost shared by all the 34 neighborhoods (above 7 and below 8), is we consider the shared letters of the LSD results in Fig 7 panel D. Finally, the rest of the mortality rate contribution to sum up to 15 (Cuauhtemoc's borough mortality description), it is distributed upon the different neighborhood model terms of Eqs (1)–(4).

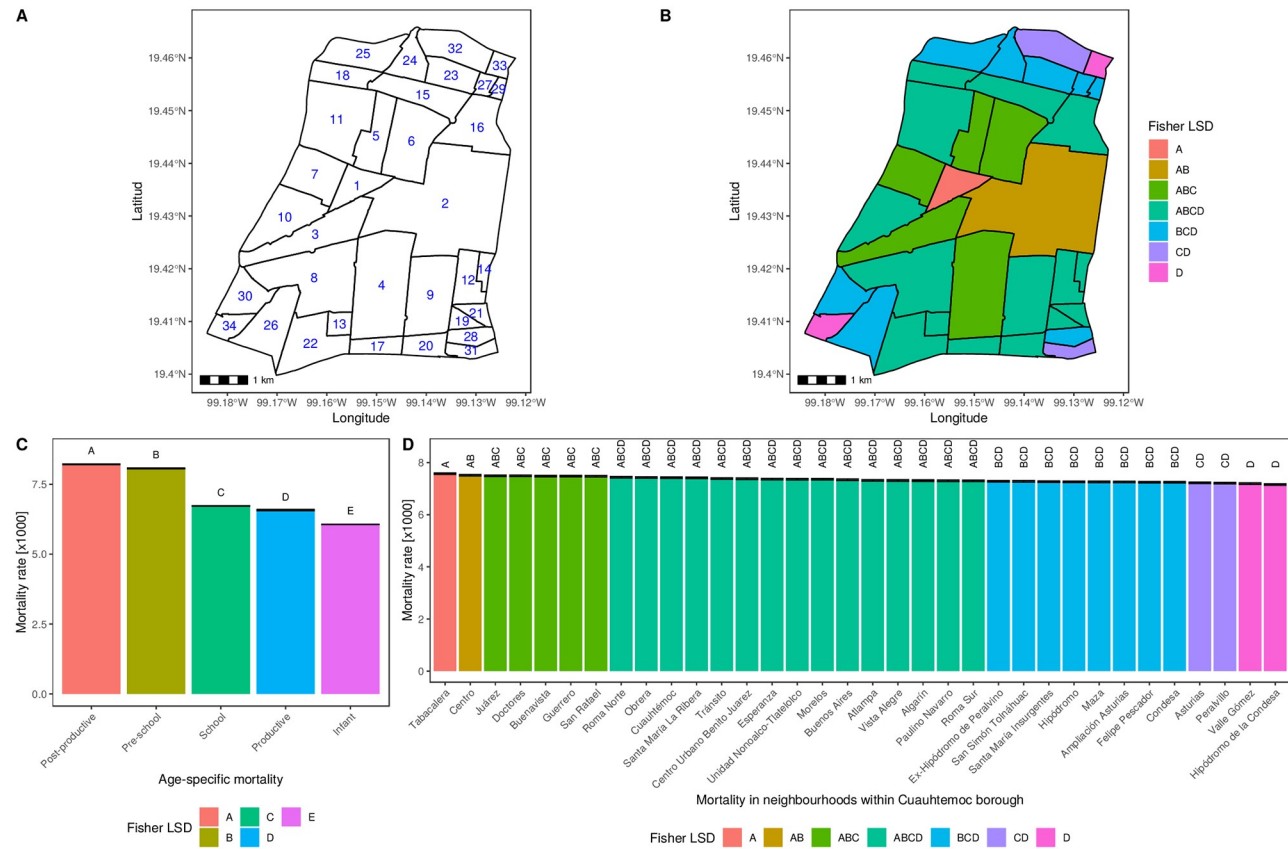

**Fig 7. Posterior neighborhood mortality test results zoom-in at Cuauhtémoc borough in Mexico City. A)** Cuauhtémoc borough is divided into its 34 neighborhoods. Numbers are ordered from the highest (1) to the lowest (34) model estimated neighborhood fixed effect mortality mean according to Eqs (1)–(4) description: 1. Tabacalera, 2. Centro, 3. Juárez, 4. Doctores, 5. Buenavista, 6. Guerrero, 7. San Rafael, 8. Roma Norte, 9. Obrera, 10. Cuauhtémoc, 11. Santa María la Ribera, 12. Tránsito, 13. Centro Urbano Benito Juárez, 14. Esperanza, 15. Unidad Hab. Nonoalco Tlatelolco, 16. Morelos, 17. Buenos Aires, 18. Atlampa, 19. Vista Alegre, 20. Algarín, 21. Paulino Navarro, 22. Roma Sur, 23. Ex-hipódromo de Peralvillo, 24. San Simón Tolnáhuac, 25. Santa María Insurgentes, 26. Hipódromo, 27. Maza, 28. Ampl. Asturias, 29. Felipe Pescador, 30. Condesa, 31. Asturias, 32. Peralvillo, 33. Valle Gómez and 34. Hipódromo de la Condesa. **B)** Neighborhoods are filled according to the Fisher's Least Significant Difference (LSD) group mean obtained at this level of representation. **C)** Age-specific Fisher's LSD results within Cuauhtémoc borough. **D)** Fisher's LSD neighborhood contribution. In all cases, capital letters stand for Fisher's LSD groups, where bars that share at least a single letter, are not statistically different after Bonferroni multiple test correction ($p > 0.05$). Results are presented as ordered mean ± standard error estimation according to model description of Eqs (1)–(4). Notice that at this data level decomposition, the LSD group means are at the same mortality rate order, i.e., the age-specific group mean values are comparable to the different neighborhood contribution. In addition, age-specific mortality values do not overlap whereas, most of the neighborhoods within this specific borough, share a common LSD group.

## Discussion

Understanding metropolitan mortality is not only relevant in terms of a mere descriptive statistical approach, but, to take into account variables that could determine, with the highest accuracy, this crucial health outcome. This knowledge, in due time, could lead to the establishment of appropriate public policies to improve citizen life quality in metropolitan areas.

In the aforementioned terms, by means of the development of a modeling approach, based on a systematic interpolation of missing data, one may observe spatio-temporal dynamics of mortality in the urban areas with higher precision. Kriging family methods have proven to be useful to achieve this goal.

In this work we have demonstrated how the improvement of the granularity level at both spatial and temporal definition, could explain some of the socio-demographic variables underlying the changes in mortality rates between boroughs. This task was achieved by statistically

interpolating those data points into a fine-grained level. In what follows, we will comment on the findings using spatio-temporal Kriging-based methods.

It can be easily noticed that Cuauhtémoc borough is the one with the highest mortality rate in all age groups, but for pre-school –which ranks in third place– as depicted in Fig 3. This borough has several particularities that should be commented, in order to unveil some hints towards plausible explanations, for the unique behavior observed there along time and space.

As observed in Fig 6C, Cuauhtémoc is the borough with the highest mortality mean value in all age groups, but in Pre-school, which shares places with Cuajimalpa and Iztapalapa boroughs. Except in that case, Cuauhtémoc has an outstanding mortality behavior over time. The case is particularly dramatic in the productive age (14 to 64 years old).

In addition, in the Productive time-evolution pattern in Fig 3 can be observed the different temporal behavior of Cuauhtémoc (solid green dashed line), compared to the rest of the boroughs. Cuauhtémoc is the economic center of Mexico City. There, the executive and legislative powers are placed, as well as the most important commerce hotspots. Indeed, Cuauhtémoc is the most densely populated borough of Mexico City.

In economic terms, Cuauhtémoc concentrates 4.6% of the gross domestic product of the entire country [75]. Around 5 million people pass through this borough every day, despite its population oscillates only around 500, 000 inhabitants. This is the place in Mexico City with the highest number of public transport stations. Cuauhtémoc also concentrates the largest markets of informal commerce of the city (Tepito Market, in the Morelos neighborhood). The high density and the flux of money and services may help to explain, to some extent, the different behavior of the mortality rate in the productive age in this borough.

Another point to take into account when mortality rate in Cuauhtémoc is observed, lies on high levels of insecurity for the aforementioned reasons regarding population density and economic concentration. Additionally, the Drug War launched at the end of 2006 by former President Felipe Calderón, affected mortality rates in a large part of the country [76], being the capital of the country also upset, in particular, the city downtown, Cuauhtémoc.

By taking into account the spatio-temporal Kriging, it has been possible to modeled mortality rates at the neighborhood level. With this spatio-temporal kriging model, the four places with the highest mortality rates were Tabacalera, Centro, Juárez and Doctores neighborhoods, part of Cuauhtémoc borough.

According to reports of the Executive Secretariat of the National System of Public Security System (Secretariado Ejecutivo del Sistema Nacional de Seguridad Pública, SESNSP), Centro is one of the most insecure boroughs of Mexico City and the homicide rate is the highest in the city [77]. By integrating transit and other accident-related issues, with employment determinants and crime-associated mortality, it may be possible to present an explanation for which the productive age mortality presents a consistent increase in Cuauhtémoc, compared to the rest of the boroughs.

Possible intrinsic biases due to possible errors in the registry must be taken into account. For example, the record of a death is registered once a certificate of death has been provided. In some cases, it could last days, depending on diverse factors. However, in the case of Mexico City, that under-registry is extremely low since social and health services are guaranteed in practically the whole area of the city.

The model presented here also has caveats, since it shows an interpolation of the coarse-grained data at the borough level of description. For instance, according to the model, Tabacalera is the neighborhood with the highest mortality rate, however, by looking at the data, the homicide rate as an example is not as large as Centro or Morelos [77].

At this stage is not possible to disambiguate whether these inconsistencies are due to the interpolating strategy or indeed reflect different causes of death, such as the ones related to

environmental factors like air pollution and other contaminants. Despite this caveat, the model shows a radial decrease of mortality in Mexico City, starting from Cuauhtémoc downtown, similar to the general behavior observed at the borough level (Fig 2).

It is relevant to notice that some of the caveats and limitations just discussed are actually instances of a well known issue known as the *modifiable areal unit problem* (MAUP). MAUP states that a bias is generated affecting statistical hypothesis testing due to the combination of two or more spatial scales on a given geostatistical analysis. The reason is that *data aggregation may become dependent on the choice of modifiable areal unit* (MAU) used as a primary source in the analysis. MAUP will in this case induces statistical biases that may lead to a form of ecological fallacy. In the particular case we presented here, since Kriging analysis is a form of regression, there are some ways to solve the MAUP or at least alleviate some of its consequences. There are a number of methodological choices to do so. We decided to follow a mixed approach by incorporating several of these as follows:

1. One of the alternatives is *to correct the variance-covariance matrix using samples from individual-level data*
   Indeed, to this first end, we have tackled the MAUP in a hierarchical manner:

   - Spatio-temporal kriging itself was modeled first, selecting the best combination of spatio-temporal variance-covariance structure (metric, separable, productSum, sumMetric and simpleSumMetric) with the appropriate single, double or triple variogram combinations (Exponential, Gaussian and Spherical), to minimize the weighted mean squared error (see Tables 2 and 3 in S1 File). Once the complete variance-covariance has been fixed, the neighborhood spatio-temporal grid was used to obtain the kriged values.

   - Over the original data, at borough level, a linear mixed-effect model using the definitions of Eqs (1)–(4). In these definitions, as in Kwok et al. 2008 [67], we did include a first-order autoregression (AR(1)) structure. In addition, data heteroscedasticity was modeled as $var(\varepsilon_{ijk}) = \sigma^2 g^2(\alpha_i)$, i.e., a multiplicative effect of the residual variance $\sigma^2$ times the variance error function $g(.)$ using a `varIdent` definition for the different age-specific mortality groups $\alpha_i$, as described in Material an Methods section.

   - Over the kriged data, at the neighborhood level, also the model of Eqs (1)–(4) was fitted to cope with the data, i.e. considering a different MAU to meet our needs.

2. A second way to correct for the MAUP is to *focus on local spatial regression rather than global regression*

3. Here, we used a global regression for the kriging process, but, considering the best variance-covariance structure possible for our data, letting the model to adjust its contribution instead of using a fixed local spatio-temporal regression. The RMSE for each mortality age-specific group is presented in Table 3 in S1 File.

4. A third strategy to correct the MAUP bias is by resorting to *design areal units to maximize a particular statistical result*. In this manuscript, we first:

   - Maximized the model likelihood while minimizing the weighted mean squared error when selecting the best spatio-temporal variance-covariance structure.

   - Maximized the linear mixed-effect model likelihood for both borough and/or neighborhood data level description.

Related approximations to solve the MAUP have been discussed in the specialized literature for decades. For instance, Raudenbush & Bryk 1986 [66] took advantage of observed unit

(individuals) when considering between and within group (in their case, schools) effects. Hence, their proposal can bee seen as a hierarchical or multilevel approach, where first, the within-group model is estimated by a separate regression equation for each school (referenced as Eq 1 in the original work). Then, the between-group model uses the regression coefficients as dependent output and try to model the within-school structural relationships (Eq 2 in [66]). Finally, the Raudenbush & Bryk can be coupled into a single equation by substitution of Eq 2 into Eq 1. The resulting equation allows to model the error term with many degrees of freedom associated with the studied variables of interest. However, the model estimation itself is not explained in detail.

Kwok et al. 2008 [67] on the other hand, resort to modeling variable time data points (not necessary at fixed intervals), unlike repeated-measurements Analysis of the Variance (ANOVA) or ANOVA polynomial trend analysis. They propose to model the variance structure by means of linear growth models using multilevel models. They start with a simple random intercept model and simple linear growth model, to show how the variance-covariance structure can tackle within-class correlation. Then, they move to models with time-invariant covariate and pseudo-$R^2$ statistic to evaluate the model's effectiveness. Finally, they present how to model covariance structure for the within-individual random errors were, the proximal autocorrelation in longitudinal data, was addressed using a first-order autoregression (AR(1)) structure.

Recent models by Anaya & Al-Delaimy 2017 [68] explicitly mention to use a "*mixed effect linear model with random effect estimation and repeated measurements*" in their work. Additionally, the hierarchical linear model in [78] and the multilevel modeling of longitudinal data in [79] were fitted using SPSS (MIXED) and SAS (PROC MIXED) procedures respectively. On the other hand, Green et al. 2019 [69] used linear mixed-effect models with R software [70] by means of `lme4` package, to fit multilevel models with random intercepts for schools and random intercepts and random slopes for time at the individual level since students are nested within waves.

These and other methods approach the MAUP from different angles. It is clear, though, that no definite nor *one-size-fits-all* solution to the problem has been found. Different problems and datasets will probably call for tailored analytical perspectives determined by their specific constraints. At the same time, further theoretical developments to the MAUP may shed some light on what kind of features a generalistic approach may have. The spatio-temporal representativeness problem (one of which instances have been discussed here) may need to be re-visited whenever further advances on how to effectively tackle the MAUP are developed.

## Conclusions

The granularity enhancement in mortality and health outcomes data will further improve its usage in a variety of public policies, such as urban development, security, creation of health centres, hospitals, public transport and even water re-usage. In these terms, efforts in gathering data at the lowest possible source (time and space), are highly desirable for future works.

Finally, in order to build predictive models that allow to tailor public policy design, the spatio-temporal dynamics of MM should incorporate socio-demographic, environmental, economical factors and relevant covariates, as detailed as possible. Future directions include, but are not limited to, utilizing environmental data, such as air pollution, quality of water, disposition of water supplies among others. Some of the above mentioned variables are actually available as open data resources. The ultimate understanding of MM by using the data presented here, as well as environmental and other risk factors, will help in the searching for the improvement of life quality of the metropolitan areas in Mexico City.

## Supporting information

**S1 File. Variogram modeling for age-specific mortality rates as well as root mean square errors for kriged data.**
(PDF)

**S2 File. Three levels of granularity for infant, pre-school and post-productive mortality rates in Mexico City.**
(PDF)

## Acknowledgments

Authors want to thank to the Subdirección de Bioinformática at the Instituto Nacional de Medicina Genómica. KBL thanks to the Consejo Mexiquense de Ciencia y Tecnología. This work is part of her PhD Thesis.

## Author Contributions

**Conceptualization:** Cristóbal Fresno, Enrique Hernández-Lemus.

**Data curation:** Karol Baca-López, Cristóbal Fresno.

**Formal analysis:** Cristóbal Fresno, Enrique Hernández-Lemus.

**Investigation:** Jesús Espinal-Enríquez.

**Methodology:** Cristóbal Fresno, Miriam V. Flores-Merino, Miguel A. Camacho-López, Enrique Hernández-Lemus.

**Project administration:** Enrique Hernández-Lemus.

**Resources:** Miriam V. Flores-Merino, Miguel A. Camacho-López.

**Software:** Karol Baca-López.

**Supervision:** Cristóbal Fresno, Jesús Espinal-Enríquez, Enrique Hernández-Lemus.

**Visualization:** Karol Baca-López, Cristóbal Fresno.

**Writing – original draft:** Karol Baca-López, Cristóbal Fresno, Enrique Hernández-Lemus.

**Writing – review & editing:** Jesús Espinal-Enríquez, Enrique Hernández-Lemus.

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
