## [Decision Letter · Decision Letter 0]

19 Jun 2020

PONE-D-20-10411

Metropolitan age-specific mortality trends at borough and neighbourhood level: The case of Mexico City

PLOS ONE

Dear Dr. Hernandez-Lemus,

Thank you for submitting your manuscript to PLOS ONE. After careful consideration, we feel that it has merit but does not fully meet PLOS ONE’s publication criteria as it currently stands. Therefore, we invite you to submit a revised version of the manuscript that addresses the points raised during the review process.

We look forward to receiving your revised manuscript.

Kind regards,

Wenjia Zhang

Academic Editor

PLOS ONE

Additional Editor Comments:

Please carefully address each comment of the two reviewers, with particular focuses on the question of 'downscale' at the neighborhood level raised by Reviewer 2 and the issues of finding clarification and language editing. Additionally, the comparison of results at borough and neighborhood levels needs to address the modifiable areal unit problem (MAUP) and more about why these two scales. For the mixed-effect linear model, while this type of models have been widely used in varying related fields, more literature reviews about them are needed to make your model analytically sound, including those similar models using different names, like multilevel modeling for longitudinal data, longitudinal multilevel model, and longitudinal HLM. You may refer to some literature below:

Raudenbush, S. W., and A. S. Bryk. A Hierarchical Model for Studying School Effects. Sociology of Education, Vol. 59, No. 1, 1986, pp. 1–7.

Kwok, O.M., Underhill, A.T., Berry, J.W., Luo, W., Elliott, T.R. and Yoon, M., 2008. Analyzing longitudinal data with multilevel models: an example with individuals living with lower extremity intra-articular fractures. Rehabilitation psychology, 53(3), p.370.

Zhang, W. and Zhang, M., 2015. Short-and long-term effects of land use on reducing personal vehicle miles of travel: Longitudinal multilevel analysis in Austin, Texas. Transportation Research Record, 2500(1), pp.102-109.

Journal Requirements:

2. We note that Figures 1, 2, 4 and 6 in your submission contain map images which may be copyrighted. All PLOS content is published under the Creative Commons Attribution License (CC BY 4.0), which means that the manuscript, images, and Supporting Information files will be freely available online, and any third party is permitted to access, download, copy, distribute, and use these materials in any way, even commercially, with proper attribution. For these reasons, we cannot publish previously copyrighted maps or satellite images created using proprietary data, such as Google software (Google Maps, Street View, and Earth). For more information, see our copyright guidelines: http://journals.plos.org/plosone/s/licenses-and-copyright.

2.1. You may seek permission from the original copyright holder of Figures 1, 2, 4 and 6 to publish the content specifically under the CC BY 4.0 license.

2.2 If you are unable to obtain permission from the original copyright holder to publish these figures under the CC BY 4.0 license or if the copyright holder’s requirements are incompatible with the CC BY 4.0 license, please either i) remove the figure or ii) supply a replacement figure that complies with the CC BY 4.0 license. Please check copyright information on all replacement figures and update the figure caption with source information. If applicable, please specify in the figure caption text when a figure is similar but not identical to the original image and is therefore for illustrative purposes only.

Reviewers' comments:

Reviewer's Responses to Questions

**Comments to the Author**

1. Is the manuscript technically sound, and do the data support the conclusions?

Reviewer #1: Yes

Reviewer #2: Partly

2. Has the statistical analysis been performed appropriately and rigorously? 

Reviewer #1: Yes

Reviewer #2: I Don't Know

3. Have the authors made all data underlying the findings in their manuscript fully available?

Reviewer #1: No

Reviewer #2: Yes

4. Is the manuscript presented in an intelligible fashion and written in standard English?

Reviewer #1: No

Reviewer #2: Yes

5. Review Comments to the Author

Reviewer #1: Major comments:

1. This paper would benefit from English editing.

2. Line 19. Mortality can be explained by individual level characteristics and population level variables. Is this what the authors mean, or do the authors want to emphasize the difference between individual level mortality (e.g. frailty) and population level mortality? I suggest rephrasing this sentence to make clearer what the authors wan to express.

3. Line 34. I don’t consider all-cause and cause-specific opposite approaches since the sum of cause-specific gives all-cause mortality.

4. The 4th (starting at line 51) and 5th paragraph seems out of place and cuts the flow of the introduction. I suggest changing this paragraph to the methodology section since is about the data and different models used.

5. There are other recent studies that also look at state-level mortality with a focus on violent deaths and other diseases that could be referenced to on cause-specific trends in Mexico City:

Gómez-Dantés, H., Fullman, N., Lamadrid-Figueroa, H., Cahuana-Hurtado, L., Darney, B., Avila-Burgos, L., ... & Aburto-Soto, T. (2016). Dissonant health transition in the states of Mexico, 1990–2013: a systematic analysis for the Global Burden of Disease Study 2013. The Lancet, 388(10058), 2386-2402.

Aburto, J. M., Beltrán-Sánchez, H., García-Guerrero, V. M., & Canudas-Romo, V. (2016). Homicides in Mexico reversed life expectancy gains for men and slowed them for women, 2000–10. Health Affairs, 35(1), 88-95.

Aburto, J. M., & Beltrán-Sánchez, H. (2019). Upsurge of homicides and its impact on life expectancy and life span inequality in Mexico, 2005–2015. American journal of public health, 109(3), 483-489.

6. Fig 4. One of the contributions of the paper is looking at more fine age groups than just looking at the all-mortality. In Fig 4, caution should be made if the rate was calculated by only dividing deaths/population*1000. Without standardizing by age-group, comparisons over time and between boroughs are inadequate. For example, the increase in the rate could be a result of the aging process and not that mortality is increasing per se. I suggest focusing on age-group mortality throughout the paper (without global) because of the mentioned reason.

7. Fig 5. It would be good to explain more about the LSD groups and what they mean.

8. Fig 6. The mortality rate shown in Fig 6 is nor consistent with Fig 5, e.g. a mortality rate of almost 8 for post productive group, and in Fig 5 is above 45. What are the explanations for this discrepancy?

9. I would like to see more discussion on data limitations. How the registry of deaths is assessed, some deaths might be counted in Mexico City even if the individual was not living there simply because there are more hospitals, etc.

10. As it is the results section is too focused on the model and less on the actual results.

11. It would also be interesting more discussion on the actual findings. Why is mortality increasing in some parts of Mexico City?

12. Is it because of health care performance? More or less hospitals? health behaviors? etc.

Minor comments:

1. Line 7. Comparing rates of what?

2. Line 12. Do you mean seasonality?

3. Line 14. Data types? Do you mean variable?

4. Line 18. Please describe which ages are included in each group since the beginning.

5. Please provide a reference to the statement ‘To worsen this situation, exposure to environmental risk factors derived from urbanization and migration has increased, with its associated negative health effects gaining attention in recent years.’

6. Line 98. These sentences are not clear.

7. Line 115. Please describe since the beginning of the study area section that you refer as Mexico City as what was previously known as D.F. and that it does not include all the Metropolitan Area.

8. Line 131. In addition, the global mortality rate group was considered, independent of the age-specific descriptor. This sentence is not clear.

9. Please consider having only one legend in Fig 2. This will allow to see how mortality is high at birth, then decreases and then increases sharply for the post-productive group.

10. Fig 3, consider reducing the size of the x-axis to more clearly see whether rates are going down or up.

11. Fig 4, consider not showing x and y labels.

Reviewer #2: This paper attempts to model metropolitan age-specific mortality at borough and neighborhood level in Mexico City. Because the data is not available at very fine spatial and temporal scales, it uses space-time kriging to get estimates at those scales. The paper then uses a time-evolution mixed effect.

The paper is overall well written, but I question the approach. First, data was only available at the borough level (which is much more coarse than the neighborhood level). Essentially, the authors were trying to fill in the gaps with space-time kriging (whichi makes sense), but they were trying to 'downscale' to obtain data at the neighborhood level; it seems to me the approach to do that is not correct. Why didn't the authors use a area-to-point kriging? Second, the authors did not provide any estimation of the error for space-time kriging, and their modeling (for instance, a root-mean square error).

Additional comments:

(1) The reader may not understand the term productive and post-productive.

(2) First sentence of the introduction? I disagree with that statement - this is not one of the most challenging. Rather write: 'a challenging'

(3) line 10: 'troubles' is not the right word. Also change 'when talking about'....this is not a formal way to write.

(4) line 20: Do you mean 'aggregated' instead of 'collectively'?

(5) line 31: Suggest you add the reference of Casas et al. (2017), which also uses a multilevel approach.

(6) line 45-50: that paragraph is not clear....need to rephrase

(7) line 55: seems a stretch to say that air pollution could explain a seasonal pattern in mortality - would be good to have more references to back this up.

(8) line 71: remove 'approach'

(9) line 75-76: The paper is not about comparing those techniques

It may be useful to add a couple of sentences describing the census in Mexico, like around lines 77.

(10) line 93-94 is an odd sentence - rephrase.

(11) line 93: 'backwardness' - what is that?

(12) line 96: change 'arise' to 'arises'

(13) sentence on line 98-101 is awkward. Rephrase.

(14) line 110 - remove 'the' before 'MM'

(15) Fig 1 - add an inset map

(16) mortality database - is the data available at neighborhood level? I guess not, and that is the point of kriging...although as I mentioned, another technique should be used

(17) We don't know the root mean square error for kriging estimates. Was a cross-validation used?

Casas, I., Delmelle, E., & Delmelle, E. C. (2017). Potential versus revealed access to care during a dengue fever outbreak. Journal of Transport & Health, 4, 18-29.

6. PLOS authors have the option to publish the peer review history of their article (what does this mean?). If published, this will include your full peer review and any attached files.

Reviewer #1: No

Reviewer #2: No

---

## [Author Response · Author response to Decision Letter 0]

2 Sep 2020

A comprehensive response to the reviewers and Academic editor has been uploaded as a PDF file. A point-by-point reponse to all coments and concerns of the reviewers as well as to the Academic Editor and Publishing editors is included in this file.

---

## [Decision Letter · Decision Letter 1]

20 Oct 2020

PONE-D-20-10411R1

Metropolitan age-specific mortality trends at borough and neighbourhood level: The case of Mexico City

PLOS ONE

Dear Dr. Hernandez-Lemus,

Thank you for submitting your manuscript to PLOS ONE. After careful consideration, we feel that it has merit but does not fully meet PLOS ONE’s publication criteria as it currently stands. Therefore, we invite you to submit a revised version of the manuscript that addresses the points raised during the review process.

We look forward to receiving your revised manuscript.

Kind regards,

Wenjia Zhang

Academic Editor

PLOS ONE

Additional Editor Comments (if provided):

Related to my previous comments, the authors provide informative responses and in-depth discussions. I appreciate their efforts. But I would suggest that they should provide a succinct discussion at the end of the paper in response to the MAUP issue (my previous comment #3) and modeling selection (my previous comment #4), by discussing existing methods and literature. Please do make some improvements before it can be acceptable. This can add two points of value to your paper. First, readers can be benefited from knowing how you select your methods (by comparing other similar methods) and how do you think about the MAUP issue. Second, it will make your paper more self-reflective by discussing some limitations or future work.

Since Reviewer #2 did not provide a further review at the sound round, I take over his role and would like the authors to address more when responding to Reviewer #2's comments, as follows:

1. I think, like above, the authors provided a good response to Comment #1. But no modification or additional information are provided in the manuscript for addressing the issues. If the reviewer had this question, possibly some readers would have similar queries.

2. Similarly, do check whether you've clarify those issues in Comments #1, 12 in your manuscript, rather than only in the response letter.

Reviewers' comments:

Reviewer's Responses to Questions

**Comments to the Author**

1. If the authors have adequately addressed your comments raised in a previous round of review and you feel that this manuscript is now acceptable for publication, you may indicate that here to bypass the “Comments to the Author” section, enter your conflict of interest statement in the “Confidential to Editor” section, and submit your "Accept" recommendation.

Reviewer #1: (No Response)

2. Is the manuscript technically sound, and do the data support the conclusions?

Reviewer #1: Yes

3. Has the statistical analysis been performed appropriately and rigorously? 

Reviewer #1: Yes

4. Have the authors made all data underlying the findings in their manuscript fully available?

Reviewer #1: No

5. Is the manuscript presented in an intelligible fashion and written in standard English?

Reviewer #1: Yes

6. Review Comments to the Author

Reviewer #1: The authors have addressed my concerns from my previous review. I have only minor changes to address in this second round of review:

Small comments:

The sentence ‘Individual mortality is influenced by personal-level characteristics such as genetics, socioeconomic status and education’ needs a reference.

The sentence ‘To worsen this situation, exposure to environmental risk factors derived from urbanization and migration has increased, with its associated negative health effects gaining attention in recent years’ is backed up with references about air pollution and urbanization, but there is no support for migration worsening health, in fact in research the evidence of this is mixed, with some cases having even a positive impact on population health. I suggest to not refer to migration as a risk factor.

In the references, the correct name of INEGI is Instituto Nacional de Estadística y Geografía.

7. PLOS authors have the option to publish the peer review history of their article (what does this mean?). If published, this will include your full peer review and any attached files.

Reviewer #1: No

---

## [Author Response · Author response to Decision Letter 1]

23 Nov 2020

A response to reviewers document has been included in the revised manuscript submission. For completeness we will copy it verbatim here.

Response to Reviewers

PONE-D-20-10411R1

Metropolitan age-specific mortality trends at borough and neighbourhood level: The case of Mexico City

PLOS ONE

Dear Professor Zhang, and Reviewers: Please find our detailed response to your editorial assessment enclosed in the present document. To ease readability our responses appear in bold blue print.

The authors are grateful to the Academic Editors and the Reviewers for their suggestions and critique of our work.

Additional Editor Comments (if provided):

Related to my previous comments, the authors provide informative responses and in-depth discussions. I appreciate their efforts. But I would suggest that they should provide a succinct discussion at the end of the paper in response to the MAUP issue (my previous comment #3) and modeling selection (my previous comment #4), by discussing existing methods and literature. Please do make some improvements before it can be acceptable. This can add two points of value to your paper. First, readers can be benefited from knowing how you select your methods (by comparing other similar methods) and how do you think about the MAUP issue. Second, it will make your paper more self-reflective by discussing some limitations or future work.

We agree with the Editor’s suggestion about addressing the MAUP in the manuscript. This discussion is now included in the revised version of the manuscript.

Since Reviewer #2 did not provide a further review at the sound round, I take over his role and would like the authors to address more when responding to Reviewer #2's comments, as follows:

1. I think, like above, the authors provided a good response to Comment #1. But no modification or additional information are provided in the manuscript for addressing the issues. If the reviewer had this question, possibly some readers would have similar queries.

2. Similarly, do check whether you've clarify those issues in Comments #1, 12 in your manuscript, rather than only in the response letter.

We agree with the Editor’s suggestion about addressing these methodological issues also in the manuscript. This discussion is now included in the revised version of the manuscript.

Reviewers' comments:

Comments to the Author

Reviewer #1: The authors have addressed my concerns from my previous review. I have only minor changes to address in this second round of review:

Small comments:

The sentence ‘Individual mortality is influenced by personal-level characteristics such as genetics, socioeconomic status and education’ needs a reference.

References have been provided to support for this statement

The sentence ‘To worsen this situation, exposure to environmental risk factors derived from urbanization and migration has increased, with its associated negative health effects gaining attention in recent years’ is backed up with references about air pollution and urbanization, but there is no support for migration worsening health, in fact in research the evidence of this is mixed, with some cases having even a positive impact on population health. I suggest to not refer to migration as a risk factor.

We have re-worded this section for clarity. We have also eliminated the reference to migration as a risk factor, as the reviewer suggested.

In the references, the correct name of INEGI is Instituto Nacional de Estadística y Geografía. 

We have corrected this in the revised version of the manuscript.

---

## [Editor Report · Decision Letter 2]

9 Dec 2020

Metropolitan age-specific mortality trends at borough and neighbourhood level: The case of Mexico City

PONE-D-20-10411R2

Dear Dr. Hernandez-Lemus,

We’re pleased to inform you that your manuscript has been judged scientifically suitable for publication and will be formally accepted for publication once it meets all outstanding technical requirements.

Kind regards,

Wenjia Zhang

Academic Editor

PLOS ONE
---

## [Editor Report · Acceptance letter]

21 Dec 2020

PONE-D-20-10411R2 

Metropolitan age-specific mortality trends at borough and neighborhood level: The case of Mexico City 

Dear Dr. Hernandez-Lemus:

I'm pleased to inform you that your manuscript has been deemed suitable for publication in PLOS ONE. Congratulations! Your manuscript is now with our production department. 

Kind regards, 

on behalf of

Dr. Wenjia Zhang 

Academic Editor

PLOS ONE